# Optimization of Process Parameters for Carbon Fiber Reinforced Polyamide 6 Composites Fabricated by Self-Resistance Electric Heating Technology

**DOI:** 10.3390/polym15081914

**Published:** 2023-04-17

**Authors:** Zhanyu Zhai, Yu Du, Xiaoyu Wang

**Affiliations:** 1College of Mechanical and Electrical Engineering, Central South University, Changsha 410083, China; 223711049@csu.edu.cn (Y.D.);; 2State Key Laboratory of Precision Manufacturing for Extreme Service Performance, Central South University, Changsha 410083, China

**Keywords:** self-resistance electric heating (SRE), mechanical properties, crystallization behavior, carbon-fiber-reinforced polyamide 6 (CFR-PA 6)

## Abstract

As an energy-saving and efficient composites-forming technology, the properties of carbon fiber self-resistance electric (SRE) heating technology still need to be improved, which is not conducive to the popularization and application of this technology. To deal with this problem, the SRE heating technology was combined with a compression molding process to form carbon-fiber-reinforced polyamide 6 (CF/PA 6) composite laminates in this study. Orthogonal experiments of three factors (temperature, pressure, and impregnation time) were designed to study the effect of process parameters on the impregnation quality and mechanical properties of CF/PA 6 composite laminates and to obtain the optimized set of process parameters. Furthermore, the effect of the cooling rate on crystallization behaviors and mechanical properties of laminates was studied according to the optimized settings. The results show that the laminates possess a good comprehensive forming quality under process parameters using a forming temperature of 270 °C, forming pressure of 2.5 MPa, and an impregnation time of 15 min. The ununiform impregnation rate is due to the ununiform temperature field in the cross-section. When the cooling rate decreases from 29.56 °C/min to 2.64 °C/min, the crystallinity of the PA 6 matrix increases from 25.97% to 37.22%; the α-phase of the matrix crystal phase also increases significantly. The effect of the cooling rate on crystallization properties also further affects the impact properties; laminates with a faster cooling rate have stronger impact resistance.

## 1. Introduction

With energy saving and emission reduction becoming the consensus of all mankind, the lightweight automobile has become an inevitable trend. Continuous carbon-fiber-reinforced thermoplastics (cCFRTs) are widely used in the automobile industry due to their advantages of high specific modulus, high specific strength, short process cycle, and good recoverability [1,2,3,4]. The traditional manufacturing processes of cCFRTs, such as compression molding [5,6,7], double-belt press technique [8,9,10], and autoclave [11,12], are based on an external heat source to heat and melt thermoplastic resin, which is accompanied by low energy utilization efficiency, causing a large amount of energy waste and low productivity.

Therefore, based on the inherent thermal and electric properties of carbon fiber, the self-resistance electric (SRE) heating method has been proposed by researchers in recent years. It is reported that the heat transfer efficiency of the SRE heating method can be over 90%, which is much higher than the 40% for traditional heating methods [13,14]. At present, there is much research on applying the SRE heating method to fabricate carbon-fiber-reinforced thermosetting resin composites, which involve electrified mode, temperature distribution, curing quality, and mechanical performance. The results of existing research suggest that, under the same process parameters, the SRE heating method has a more significant effect of energy saving [15,16], more uniform temperature distribution [16,17], and can achieve similar curing quality and mechanical properties compared with those conventional forming methods [18,19,20,21]. As for forming cCFRTs by SRE heating, Enoki et al. [22,23] studied the SRE heating response of carbon fiber no-crimp fabrics in two-dimensional shapes and three-dimensional shapes and observed the impregnation quality of CF/PA 6 laminates formed by the SRE heating method combined with compression molding. The results indicate that excellent forming quality can be achieved by this method, and the temperature in the curved area of carbon fiber with a three-dimensional shape is significantly lower than in other areas. Reese et al. [24] developed a novel SRE heating method that applied a current on the dry hybrid yarn textile with recycled carbon fibers and PA 6 fibers in the transverse fiber direction. This method allows the completion of the impregnation and consolidation process in less than one minute and achieves the flexural properties of composites fabricated by this method compared to existing processes. However, the existing research about the application of the SRE heating method on cCFRTs is still limited and unsystematic. Our previous research has demonstrated that the compression molding process with SRE heating is comparable to or even superior to the conventional compression molding process by comparing the surface chemical properties, crystallization, impregnation quality, and mechanical properties for cCFRTs formed by the two processes [25]. Nevertheless, the impregnation quality and mechanical properties of CFRTs formed by this process still have great space for improvement. 

The process parameters have been proven to have a direct and significant effect on the forming quality [26]; in addition, the impregnation quality and mechanical properties of cCFRTs can be improved simply and effectively by adjusting the process parameters. Lu et al. [27] fabricated continuous carbon-fiber-reinforced polyether-ether-ketone (PEEK) composites through hot compression and studied how the process parameters, including melting temperature, crystallization temperature, molding pressure, and resin content, affect the properties of CF/PEEK composites. Ishida et al. [28] investigated the effect of pressure on the flow and impregnation behavior of resin under the double-belt press process of cCFRTs. The results indicate that the resin has two flow modes of in-plane flow and impregnation due to the pressure gradient in and out of the rolls; in addition, that the impregnation quality of cCFRTs is affected by the combination flows. Furthermore, when the SRE heating method is introduced, the temperature distribution in the plane area and thickness direction of carbon fabric changes significantly [29], which would influence the resin flow during processing. On the other hand, many researchers have demonstrated that the crystallization of semi-crystalline thermoplastics is strongly dependent on process conditions, especially on the cooling rate. Taketa et al. [30] statistically evaluated the influence of the cooling rate on the strength of unidirectional thermoplastic prepreg systems (et al. CF/PA 6, CF/PP, and CF/PPS). They found that, affected by the change in the degree of crystallinity, the toughness of the CF/PA 6 matrix decreased along with the decrease in the degree of the cooling rate. To the best of our knowledge, very few studies have been performed to study the effects of process parameters (i.e., temperature, pressure, impregnation time, and cooling rate) on the impregnation behavior and mechanical properties of cCFRTs fabricated by the compression molding process combined with the SRE heating method.

In this paper, SRE heating combined with the compression molding process was applied to form carbon-fiber-reinforced polyamide6 composite laminates. Initially, an orthogonal experiment was designed to investigate the effect of process parameters (i.e., temperature, pressure, and impregnation time) on forming quality. To evaluate the forming quality, the porosity and mechanical properties of obtained CF/PA 6 composite laminates were measured and discussed. Then, the optimized process parameters of each property were summarized. Especially, based on the obtained optimal process parameters; the influence of the cooling rate on crystallization properties and mechanical properties was also investigated.

## 2. Experiments

### 2.1. Used Materials

The material chosen as reinforcement for this study was plain weave carbon fiber fabric (CO6343B, Toray Co., Ltd., Tokyo, Japan). Polyamide 6 (PA 6) film (Ultramid^®^ B3S, BASF Co., Ltd., Ludwigshafen, Germany) with 0.3 mm thickness was chosen as the polymer matrix for CF/PA 6 composite laminate. Before CF/PA 6 composite laminate manufacturing, the PA 6 films were pre-dried at 80 °C for 12 hours to remove the excess moisture.

### 2.2. Manufacturing of CF/PA 6 Composite Laminate

The film stacking technique was chosen for the manufacturing of CF/PA 6 composite laminate, which is based on self-developed compression molding with the SRE heating system, as shown in Figure 1. The forming process was carried out on a hot compressing machine (HBSCR-30T/400A, HUABO MACHINERY Co., Ltd., Wenling, China). The initial mold temperature was controlled by a self-developed temperature-controlling device. The SRE heating process was achieved using a 1080 W DC power supply (2260B-30-108, Tektronix Inc Co., Ltd., Beaverton, OR, USA) connected to the carbon fiber fabrics with wires and copper electrodes.

Two plies of carbon fabric and three plies of PA 6 film were stacked alternately. The sizes of the carbon fabric and PA 6 films were 130 mm × 90 mm and 100 mm × 100 mm, respectively. The stacking material was placed into the mold mounted on the hot compressing machine. Herein, the stacking material was wrapped with thermal insulation material to reduce the loss of joule heat and improve the uniformity of process temperature. During the SRE heating process, electric power was applied to the carbon fabric through the length direction. The manufacturing process for CF/PA 6 composite laminate includes: (1) closing the mold and heating it to 80 °C; (2) applying the electric current on carbon fabric under a specific pressure; (3) impregnating the carbon fabric with molten PA 6; (3) cutting off the electric power after a specific time and cooling the CF/PA 6 laminate to 80 °C; (4) releasing the pressure and demolding. The process parameters were chosen based on much research and pre-experiments, as described in Section 2.3.

### 2.3. Determination of Process Parameters

According to the previous DSC test results [27], the melting temperature (Tm) of PA 6 film is 221.3 °C. Research by Mayer et al. [10] suggests that the suitable forming temperature for the film stacking technique is in the range of 1.1~1.3 Tm. Additionally, the recommended forming temperature for the PA 6 film is in the range of 250~270 °C given by BASF. Therefore, the forming temperature for CF/PA 6 composites was selected as 250 °C, 260 °C, and 270 °C. The forming pressure was determined by the pre-experiments. CF/PA 6 composite laminates were manufactured under the processing temperature of 270 °C and the impregnation time of 20 min with different forming pressures of 1.0 MPa, 1.5 MPa, 2.0 MPa, 2.5 MPa, and 3.0 MPa. The cross-section morphologies of laminate fabricated under different forming pressure were observed, as shown in Figure 2. It can be seen that the carbon fibers could be fully impregnated under the forming pressure of 2.0 MPa and 2.5 MPa, and there were a few carbon fibers that fractured when the pressure reached 3.0 MPa. Therefore, the forming pressures were selected as 1.0 MPa, 1.5 MPa, 2.0 MPa, and 2.5 MPa. As for impregnation times, they were selected as 5 min, 10 min, 15 min, and 20 min, based on references [31,32]. Without affecting the reliability of research results, an orthogonal experimental plan was designed to reduce the workload, as shown in Table 1.

### 2.4. Determination of Cooling Rate

Based on the control of current during SRE heating, a set of cooling rates was obtained. During the cooling, instant power-off, current reduction by 3 A/min, and reduction by 0.6 A/min were used, respectively. When the temperature of the laminate matched that of the mold (approximately 100 °C), the mold was opened and the laminate was removed. The embedded location of thermocouples (TT-K-40, CESMOOY, Shanghai, China) was shown in Figure 3. The Points T1–T4 were, respectively, contacted to centers of the upper and lower surfaces of each fabric layer. Points T5–T6 were located along the diagonal of the upper surface of the second layer of fabric. The measures were repeated three times. To eliminate the error, the average temperatures were calculated. Figure 4 shows the changes in temperature during cooling. The cooling rates measured were 29.56 °C/min, 10.87 °C/min, and 2.67 °C/min, respectively.

### 2.5. Characterizations

#### 2.5.1. Currents and Corresponding Temperature Uniformity

The currents for different processing temperatures were measured and the corresponding temperature distribution of the CF/PA 6 composite laminates was recorded simultaneously by thermocouples (TT-K-40, CESMOOY, Shanghai, China) embedded into the different locations, as seen in Figure 3. The temperature measured at T3 was used as the reference to determine the current. In order to improve the heating rate, a high current of 60 A was applied first, then the current was gradually decreased to stabilize the temperature after the carbon fiber fabrics reached the target temperature. The moments of each reduction in current were recorded in order to manufacture CF/PA 6 composite laminates under the totally same parameters subsequently. The tests were repeated three times and the average temperatures were calculated to eliminate the error.

#### 2.5.2. Impregnation Quality

According to the standard of ASTM D2734 [33], the void contents of composite laminates manufactured under different processing parameters were calculated based on the following equations,
(1)$$\rho_{ct}=100/\left(W_{r}/\rho_{r}+W_{f}/\rho_{f}\right)$$
where are the resin mass fraction of composite, the fiber mass fraction of composite, theoretical composite density, the density of PA6, and the density of carbon fiber, respectively;
(2)$$V_{v}=100\left(\rho_{ct}-\rho_{c}\right)/\rho_{ct}$$
where $V_{v}$ and $\rho_{c}$ are the porosity of composite laminate and the measured density of composite laminate, respectively. The $\rho_{r}$, $\rho_{f}$, and $\rho_{c}$ were measured by the buoyancy in a fluid with known density which utilizes the Archimedes principle. The $W_{r}$ and $W_{f}$ were measured by definition of the mass fraction, in which a piece of carbon fiber fabric with 50 mm × 50 mm and its resulting laminate were weighted.

#### 2.5.3. Crystallization Properties

XRD analyses were carried out using an X-ray Diffraction Instrument (D8 Advance, Bruker, Germany) to study the phase and crystal structure of composites. By using a water-cooled sawing machine, the test samples were cut into 20 mm × 20 mm from the center of the laminates fabricated with different cooling rates. After the experiment, the obtained X-ray diffraction spectrum was fit to analyze the crystallinity and phase content of the sample.

#### 2.5.4. Mechanical Properties Tests

An electronic universal testing machine (CMT4204, MTS Co., Ltd., Eden Prairie, MN, USA) was used to carry out the flexural tests based on the standard of ASTM D7264/D7264M [34] with a loading rate of 1.0 mm/min. The flexural test samples were cut from the laminates manufactured under different processing parameters with dimensions of 25 mm × 13 mm × h, and the span thickness ratio was selected to be 20:1. The h is the inherent thickness of laminates, measured to be 0.9 ± 0.05 mm. Five samples were tested to eliminate the error per processing parameter. The concentrated quasi-static indentation (QSI) tests were performed according to the standard of ASTM D6264/D6264M [35] on an electronic universal testing machine (E44, MTS Co., Ltd., Eden Prairie, MN, USA). The loading rate was set as 1.25 mm/min, and the dimensions of QSI testing samples were 50 mm × 50 mm × h.

#### 2.5.5. Microscopic Morphology Observation

The cross-sectional micrographs of the CF/PA 6 laminates were observed by a digital microscope (DSX10-UZH, Olympus Co., Ltd., Tokyo, Japan). The damaged area of samples after QSI tests was taken by a camera.

## 3. Results and Discussion

### 3.1. Currents and Corresponding Temperature Uniformity

The input currents required for the three forming temperatures (250 °C, 260 °C, and 270 °C) applied in this study are shown in Figure 5a. After heating to the target temperature with an initial current of 60 A, the current needs to be gradually reduced to 40.5 A, 43.0 A, and 47.0 A, respectively, to maintain temperature stability. Figure 5b–d shows the temperature response curves at different positions under different input currents in the whole SRE heating process. It can be observed in Figure 5 that the temperature can be raised to the target temperature in 80 s and be kept stable as the current decreases. The steady temperatures of different measuring positions are shown in Figure 5e,f. The results show that the highest temperature in the thickness direction is T3, which is on the upper surface of the second carbon fiber fabric layer. As the forming temperature increases from 250 °C to 270 °C, the temperature difference in the thickness direction increases from 20 °C to nearly 40 °C. The increase of temperature difference is mainly due to the lowest temperature at the T4 point, which does not increase significantly with the increase of temperature. It is because the mold temperature is only 80 °C and the temperature gradient between the carbon fiber fabric and the mold is nearly 200 °C, which leads to intense heat dissipation at the T4 point, the nearest point to the mold. Therefore, although the temperature at the T3 point increases by 20 °C, there is no obvious temperature change at the T4 point. The temperature difference via the in-plane direction is about 20 °C and does not change significantly with the increase in temperature.

### 3.2. The Effect of Process Parameters on Forming Quality

In this section, we will first report all the orthogonal experiment results, as shown in Table 2. The effects of process parameters on the impregnation quality, flexural properties, and impact properties of CF/PA6 composite laminates formed by the SRE heating method will be analyzed in the following sections. In order to investigate the significance of the effects of the process parameters, an analysis of variance (ANOVA) was carried out with the measured void contents, flexural properties, and impact properties. In ANOVA, the F-test was used to analyze whether the effect of each factor was significant, and the *p*-value could show the order of significance for the process parameters.

#### 3.2.1. The Effect of Process Parameters on Void Content

The ANOVA results indicate that the three process parameters (molding temperature, molding pressure, and impregnation time) have significant effects on void contents (*p* < 0.05). The order of significance is molding pressure (*p* = 0.000300) > molding temperature (*p* = 0.000352) > impregnation time (*p* = 0.000416), under the process parameters investigated. Although the significance of the three process parameters is different, the *p* values of all three process parameters are less than 0.001, which indicates that the three process parameters are strongly correlated with the void content of CF/PA6 composite laminates. The optimal set of process parameters for reducing void content is determined according to the line chart of estimated values, as shown in Figure 6. It can be seen from the figure that the void content decreases first and then increases with the increase in temperature and impregnation time, and the void content decreases with the increase in pressure. The adjustment of these three process parameters can effectively reduce the void content. After a comprehensive analysis, the optimal set of process parameters means a molding temperature of 270 °C, a molding pressure of 2.0 MPa, and an impregnation time of 20 min.

Four groups of the laminates, formed by process parameters with Exp.ID of 3, 10, 9, and 11 (the corresponding void contents are 9.16%, 5.60%, 2.39%, and 0.62%), were selected to observe the impregnation morphology of the section, analyze the evolution trend of impregnation morphology, and verify the accuracy of the porosity calculations, as shown in Figure 7. When the void content is 9.16%, there are many dry fibers in the fiber bundles. When the void content is 5.60%, there are only microvoids in most of the fiber bundles, and there are many dry fibers unimpregnated in a few fiber bundles. When the void content is 2.39%, parts of the fiber bundles have been impregnated completely, and there are only a few microvoids in most of the fiber bundles. When the void content is 0.62%, the majority of fiber bundles are completely impregnated, and only a small number of fiber bundles have microvoids.

#### 3.2.2. The Effect of Process Parameters on Flexural Properties

According to the F-test results, the molding temperature, molding pressure, and impregnation time all have significant effects on the flexural strength of the laminates (*p* < 0.05). While, the *p*-value of the molding pressure for flexural modulus is greater than 0.05, indicating that the molding pressure has no significant effect on the flexural modulus. According to the *p*-value, the order of significance is molding temperature > impregnation time > molding pressure. Figure 8a shows that the flexural strength increases with the increase of forming temperature and impregnation time. When the forming pressure increases, the flexural strength of CF/PA 6 laminates increases first and then decreases. When the pressure is 2.0 MPa, the flexural strength is the highest. The corresponding optimal process parameter set is a molding temperature of 270 °C, a molding pressure of 2.0 MPa, and an impregnation time of 20 min. As for the flexural modulus, it generally increases with the increase of temperature, pressure, and impregnation time, as shown in Figure 8b. Therefore, the optimal process parameter set of the flexural modulus is a molding temperature of 270 °C, a molding pressure of 2.5 MPa, and an impregnation time of 20 min. In view of the insignificant effect of forming pressure on the flexural modulus, the optimal set of process parameters for flexural strength and modulus is a forming temperature of 270 °C, a forming pressure of 2.0 MPa, and an impregnation time of 20 min.

#### 3.2.3. The Effect of Process Parameters on Impact Properties

The QSI test reflects the impact properties through the two key indicators; one is the peak force when the laminates are first damaged, and the other is the energy absorbed by the laminates when they are completely damaged. Figure 9 shows the load–displacement curves and the energy absorption curves of the laminates in the QSI tests. It can be seen that the laminate mainly went through three stages during every test. Firstly, the load gradually increased to the peak. Then minor damages occurred continuously to absorb energy. At this point, the loading was stable with slight fluctuations. Finally, the laminate was completely penetrated, and the test was over. According to the results of the ANOVA, for the maximum indentation force, the *p*-value of each three process parameters is greater than 0.05. For the energy absorption value, only the *p*-value of the impregnation time is less than 0.05, which indicates that the effect of the process parameters on the impact performance is not significant. This may be due to the fact that the impact properties are mainly affected by the crystallization properties of the matrix, while the three process parameters studied in this research mainly relate to the impregnation quality and have little effect on the crystallization properties. Therefore, the effects of forming temperature, forming pressure, and impregnation time on impact properties are not considered in the optimization of process parameters.

In terms of impregnation quality, the optimal set of process parameters is 270 °C, 2.5 MPa, and 15 min; while in terms of flexural properties, the optimal set of process parameters is 270 °C, 2.0 MPa, and 20 min. The orthogonal experiment results show that the laminates formed by the two sets of process parameters are similar in flexural properties. However, the impregnation time of 15 min is more conducive to reducing energy consumption and improving efficiency. Therefore, the optimal set of process parameters is selected as a forming temperature of 270 °C, a forming pressure of 2.5 MPa, and an impregnation time of 15 min. In addition, the void content of the laminates formed by this set of process parameters is only 0.62%, which fully meets the requirements of composite materials for the automotive industry and even the aerospace industry.

### 3.3. The Effect of Cooling Rate on Properties

#### 3.3.1. The Effect of Cooling Rate on Crystallization Behavior

Figure 10 presents the XRD test diffractograms which intercepted 2θ in the range of 18°~28°. The samples prepared under three different cooling rates all have obvious characteristic peaks at 2θ = 20°~24°, which the corresponding crystal phase composition is the α-phase. However, there is no obvious characteristic peak at 2θ = 21°, which corresponds to the crystal phase composition of the γ-phase. The results highlight that the main crystal phase composition of the sample PA 6 matrix formed under different cooling rates is the α-phase; in which, molecular chains are tightly entangled and the crystal structure is relatively stable, rather than the unstable γ-phase.

The content of the crystal phase can be reflected by the area of the diffraction characteristic peak. It can be seen that the content of the α-phase of the sample increases gradually when the cooling rate decreases gradually, as shown in Figure 10b. It is because the cooling rate decreases, making the overall temperature of the crystallization process higher and the crystallization time more sufficient; hence, the molecular mobility gradually increases and the entanglement between molecular chains is more likely to form a stable crystal structure. Similarly, the crystallinity of the sample also Increases gradually with the decrease of the cooling rate, as shown in Figure 10c; when the cooling rate decreases from 29.56 °C/min to 2.64 °C/min, the crystallinity of the PA 6 matrix increases from 25.97% to 37.22%.

#### 3.3.2. The Effect of Cooling Rate on Impact Properties

The results of laminates manufactured at different cooling rates carried out by the QSI test under 25 °C and 100 °C were obtained, as shown in Figure 11. The maximum failure peak force of the laminates under 25 °C and 100 °C environments decreased with the decrease of the cooling rate, by 23.22 N and 32.79 N, respectively, as shown in Figure 11a. The total energy absorbed by the sample, which was manufactured at different cooling rates, after being completely destroyed under different test temperatures, was shown in Figure 11b. Under the same test temperature, the energy absorption values of the samples decreased with the decrease of the cooling rate, by 0.21 J and 0.28 J, respectively. This is because with the decreasing of the cooling rate, the crystallinity and the α-phase content of the PA 6 matrix increase. The ductility of the α-phase is relatively low [36], which weakens the impact resistance of the PA 6 matrix. Furthermore, the fully crystallized crystals around the fibers and the decreases in the cooling rate result in low-impact, load-bearing capacity between fibers and the matrix. Compared with the normal temperature, the peak force and energy absorption value of laminates in the high temperature have been improved overall. This may be due to that in the high temperature, the migration and diffusion ability of the molecular chains in the grains of the PA 6 matrix is enhanced, which is manifested in the enhancement of the toughness at the macro level; therefore, improving the impact performance of the laminate. As shown in Figure 11c, the energy absorption during the test can be mainly divided into three stages. First, the load increases gradually until reaching the peak force and failure occurs; at this stage, the slope of the energy absorption curve gradually increases. Second, when the load reaches the peak force, the laminate absorbs energy by continuing to occur microdamage. In this stage, the load enters a stable plateau period with slight fluctuation. Third, the laminate is completely destroyed and penetrated. At this stage, the load is mainly created by the friction between the laminate and the indenter, which decreases as time goes by, causing the slope of the energy curve to decrease gradually. Figure 11d shows the change in the ratio of energy absorption with the cooling rate. The ratio of energy absorption was calculated by dividing the energy absorbed at 100 °C by that at 25 °C. The result indicates that the improvement of the high temperature to the performance of the laminate becomes higher as the cooling rate decreases. From the destruction morphology of each sample after the QSI test, as shown in Figure 12, all of the surfaces of the laminates exhibited crisscross cracks. Compared with the high temperature, the laminate with a cooling rate of 2.64 °C/min at normal temperature has more damage and cracks, which can be attributed to high crystallinity and low toughness due to the low cooling rate.

## 4. Conclusions

In this paper, the effects of process parameters on the impregnation quality and mechanical properties of carbon-fiber-reinforced thermoplastic composite laminates were investigated. To find the suitable process window during the compression stage, the influences of the process parameters on forming quality were investigated through experimental characterization. The void content, flexural properties, and impact properties of obtained laminates were discussed to evaluate the forming quality. Furthermore, based on the optimized set, the effect of the cooling rate on crystallization behavior and impact properties was investigated by XRD analysis and the QSI test.

The main conclusions from the experimental results are as follows.

The increasing of parameters (et al., molding temperature, molding pressure, and impregnation time) favors matrix flowing and fiber impregnation, leading to lower void content. However, high molding temperature and long impregnation time can weaken the impregnation effects on laminates. On the other hand, increasing three parameters contributes to higher flexural properties, including flexural strength and flexural modulus, but shows insignificant effects on impact properties. Under this technological process, the optimal set of process parameters is selected as a forming temperature of 270 °C, a forming pressure of 2.5 MPa, and an impregnation of 15 min.The XRD analyses reveal that with the cooling rate decrease from 29.56 °C/min to 2.64 °C/min, the crystallization of PA 6 laminates improves from 25.97% to 37.22%. This trend corresponds with the variation trend of α-phase content in the matrix.The peak force of the sample at the test temperature of 25 °C and 100 °C is decreased by 23.22 N and 32.79 N, respectively, with the decrease in the cooling rate. Moreover, both reducing the cooling rate and raising the test temperature can have a positive effect on the energy absorbed by the sample. It can be obtained that the lower cooling rate contributes to poorer impact properties, while the higher test temperature in the QSI can improve impact properties.

## Figures and Tables

**Figure 1 polymers-15-01914-f001:**
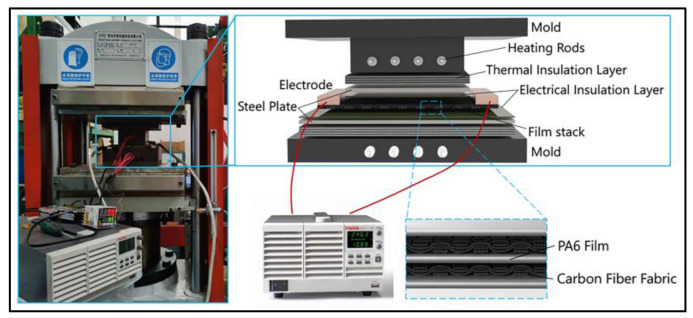
The photo image and schematic drawing of the hot compression mold for CF/PA 6 composite laminate with the SRE heating system.

**Figure 2 polymers-15-01914-f002:**
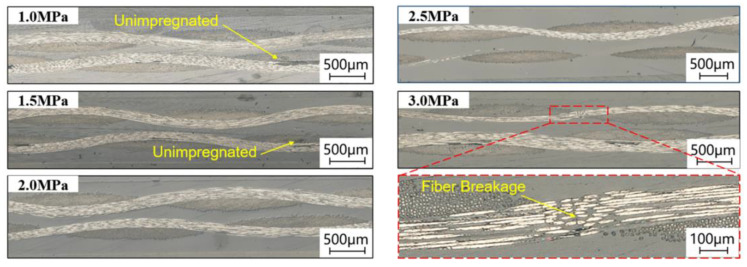
Micrographs of cross-section morphologies of CF/PA 6 composite laminates formed under different pressures.

**Figure 3 polymers-15-01914-f003:**
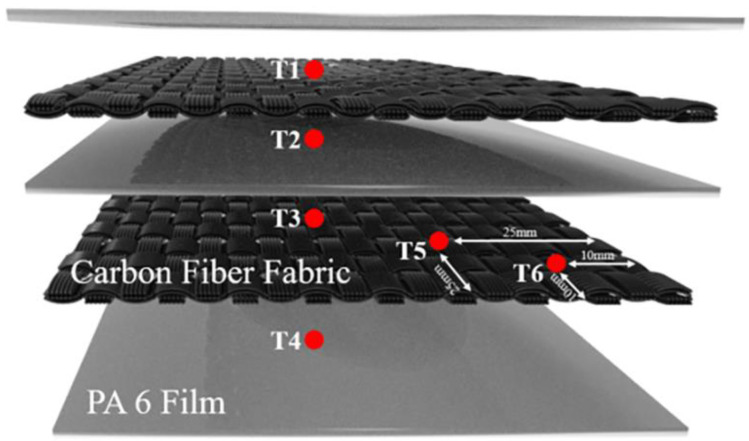
Schematic of locations of embedded thermocouples in a stack CF/PA 6 composites laminate under SRE heating.

**Figure 4 polymers-15-01914-f004:**
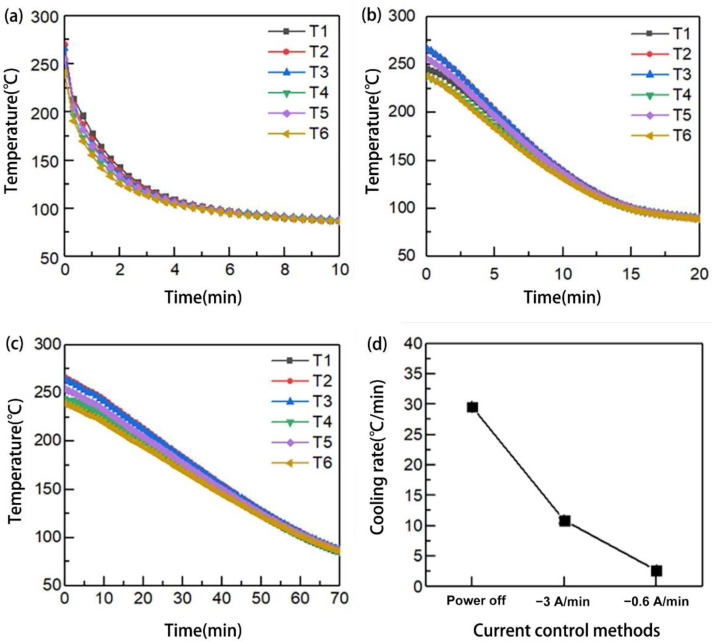
Currents control method and temperature changes: (**a**) temperature curve during cooling by power-off; (**b**) temperature curve during cooling by current reduction of 3 A/min; (**c**) temperature curve during cooling by current reduction of 0.6 A/min; (**d**) cooling rate under different current control methods.

**Figure 5 polymers-15-01914-f005:**
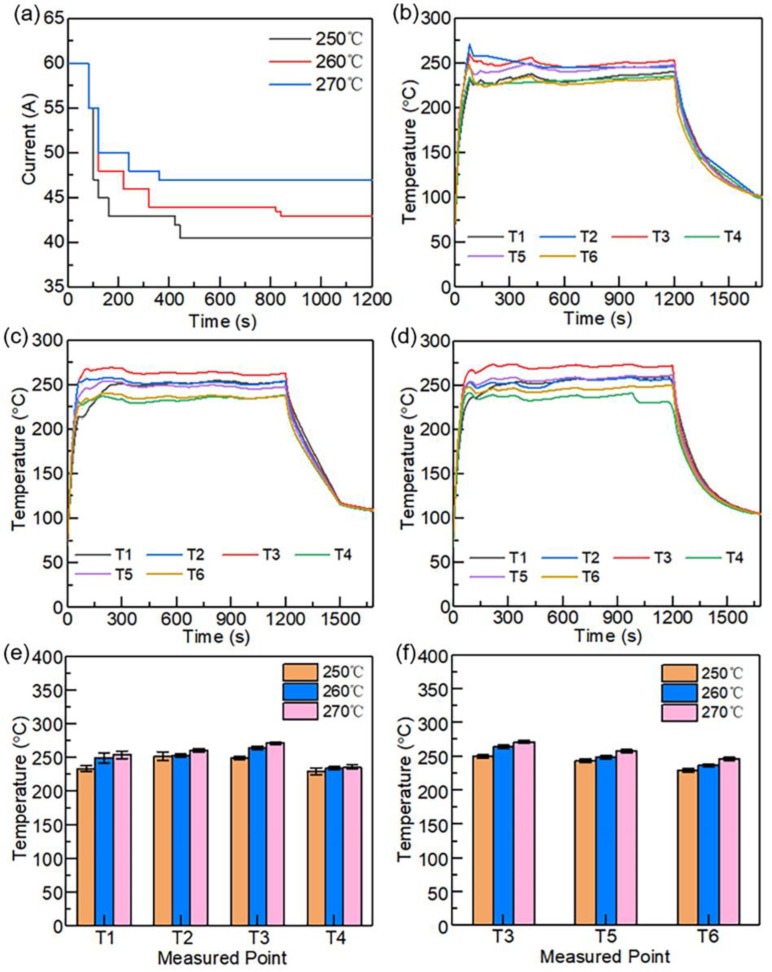
Input currents and corresponding temperature responses: (**a**) input current required for different forming temperatures; (**b**) heating curves at different positions under the forming temperature of 250 °C; (**c**) heating curves at different positions under the forming temperature of 260 °C; (**d**) heating curves at different positions under the forming temperature of 270 °C; (**e**) average temperatures at different positions along the thickness direction; (**f**) average temperatures at different positions along the in-plane direction.

**Figure 6 polymers-15-01914-f006:**
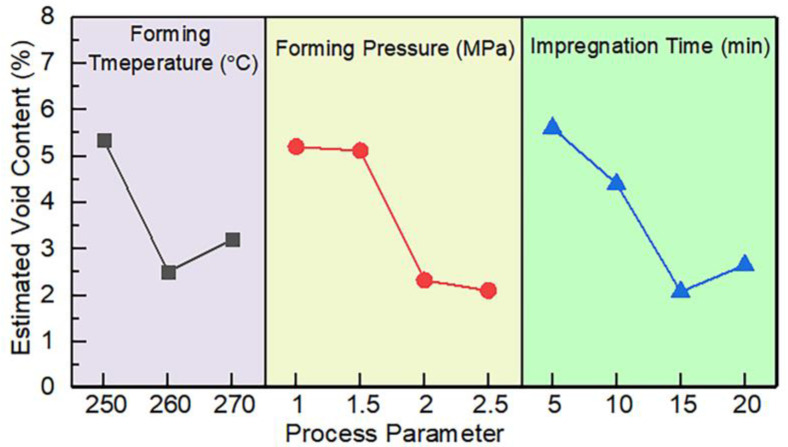
The effect of process parameters on the void content.

**Figure 7 polymers-15-01914-f007:**
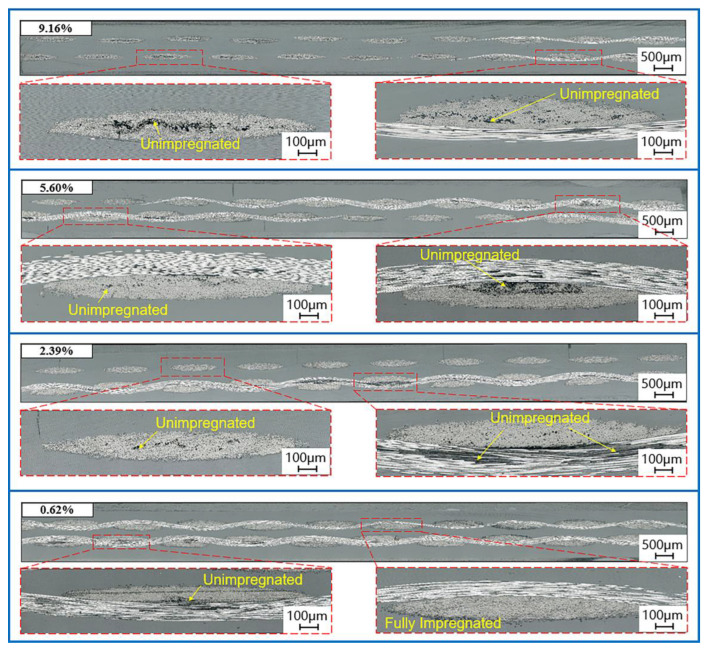
Section impregnation morphology of laminates with different void contents.

**Figure 8 polymers-15-01914-f008:**
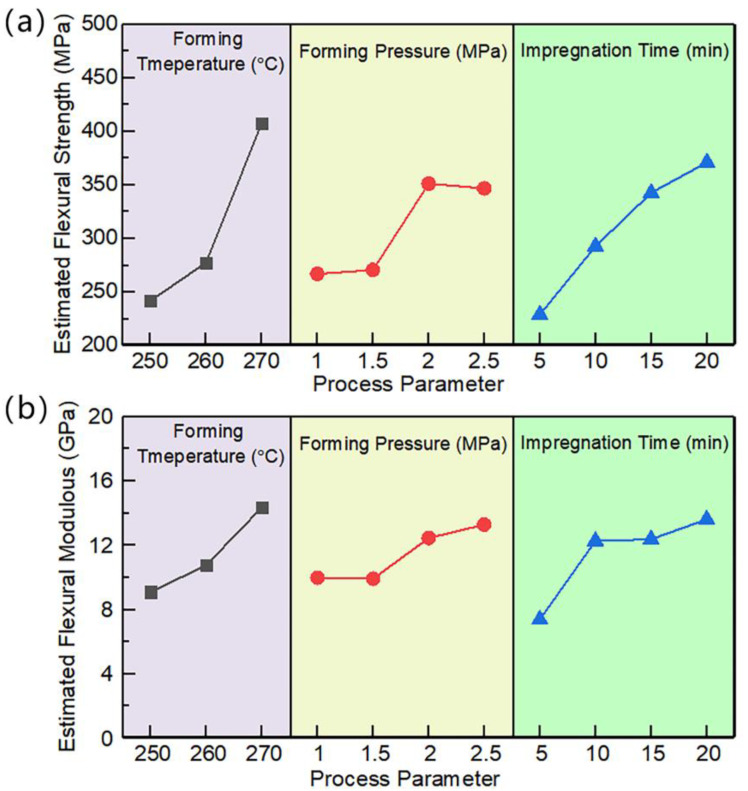
The effect of process parameters on the flexural properties: (**a**) flexural strength; (**b**) flexural modulus.

**Figure 9 polymers-15-01914-f009:**
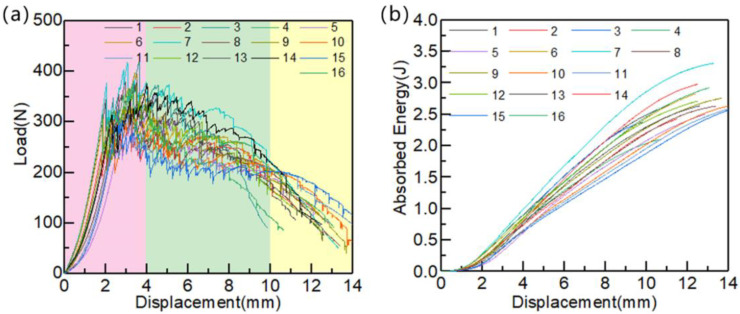
The QSI test results of CF/PA 6 composite laminates formed under different process parameters: (**a**) load–displacement curve; (**b**) energy absorption curve.

**Figure 10 polymers-15-01914-f010:**
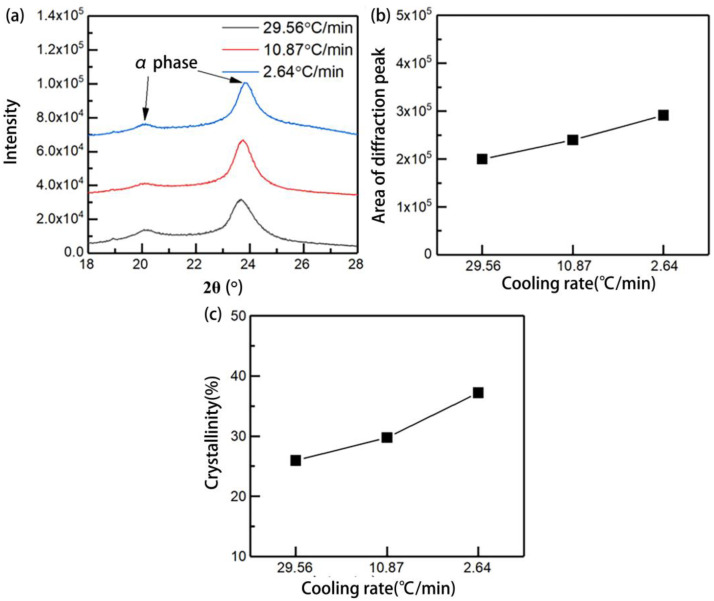
XRD results for different cooling rates: (**a**) XRD test diffractograms; (**b**) area of diffraction peak; (**c**) crystallinity of the sample.

**Figure 11 polymers-15-01914-f011:**
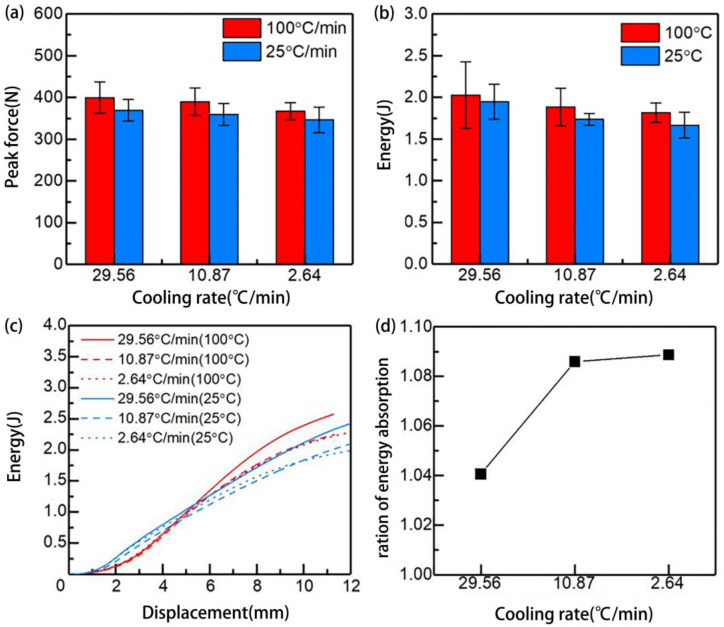
The results of laminates manufactured at different cooling rates carried out by the QSI test under 25 °C and 100 °C: (**a**) peak force when failure occurred; (**b**) total energy absorbed by the sample; (**c**) energy curve; (**d**) ratio of energy absorption.

**Figure 12 polymers-15-01914-f012:**
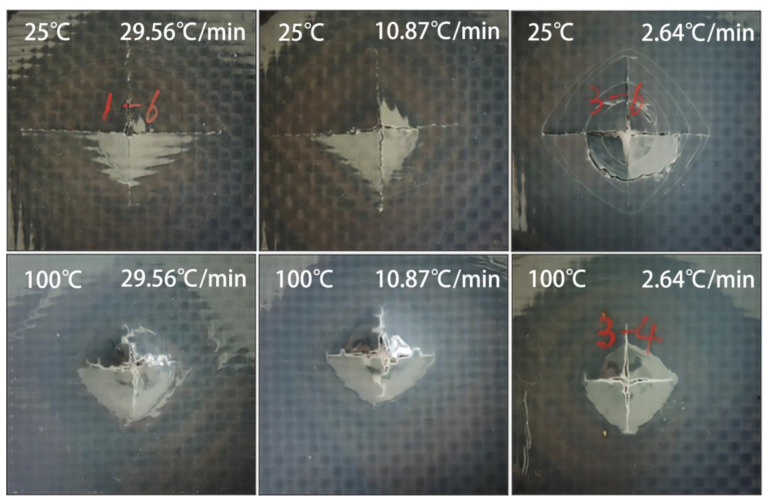
Destruction morphology of a sample of each cooling rate after the QSI test.

**Table 1 polymers-15-01914-t001:** Summary of the process parameters used for the manufacturing of CF/PA 6 composite laminate.

Exp. ID	Processing Temperature (°C)	Forming Pressure (MPa)	Impregnation Time (min)
1	250	2.5	20
2	260	1.0	20
3	250	1.0	5
4	270	1.5	5
5	250	2.0	5
6	260	2.5	5
7	250	2.0	15
8	250	1.5	20
9	260	1.5	15
10	270	1.0	10
11	270	2.5	15
12	250	2.5	10
13	250	1.5	10
14	250	1.0	15
15	270	2.0	20
16	260	2.0	10

**Table 2 polymers-15-01914-t002:** Void contents, flexural properties, and impact properties of CF/PA 6 composite laminates.

Exp. ID	Process Parameters	Void Content (%)	Flexural Properties		Impact Properties	
			Flexural Strength (MPa)	Flexural Modulus (GPa)	Maximum Indentation Force (F)	Absorbed Energy (kJ)
1	250 °C/2.5 MPa/20 min	2.44	243.66	13.88	378.32	2.21
2	260 °C/1.0 MPa/20 min	3.47	325.08	13.32	304.11	1.99
3	250 °C/1.0 MPa/5 min	9.16	153.60	3.67	419.14	1.86
4	270 °C/1.5 MPa/5 min	6.38	271.58	10.37	368.44	1.72
5	250 °C/2.0 MPa/5 min	6.19	209.23	5.86	356.76	2.03
6	260 °C/2.5 MPa/5 min	2.37	215.46	7.34	336.81	2.27
7	250 °C/2.0 MPa/15 min	2.72	182.12	13.4	419.06	2.38
8	250 °C/1.5 MPa/20 min	6.14	245.32	9.33	354.10	2.13
9	260 °C/1.5 MPa/15 min	2.39	270.41	9.74	396.44	2.37
10	270 °C/1.0 MPa/10 min	5.60	308.81	14.02	307.70	2.17
11	270 °C/2.5 MPa/15 min	0.62	519.87	17.47	38	2.22
12	250 °C/2.5 MPa/10 min	4.64	267.76	12.13	373.94	2.07
13	250 °C/1.5 MPa/10 min	7.22	229.00	7.91	382.59	2.10
14	250 °C/1.0 MPa/15 min	4.23	213.32	6.57	375.37	2.41
15	270 °C/2.0 MPa/20 min	0.23	529.55	15.58	320.22	2.07
16	260 °C/2.0 MPa/10 min	1.81	299.09	12.67	366.94	2.27

## Data Availability

The data presented in this study are available on request from the corresponding author.

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
