# Peer review of "Optimization of Process Parameters for Carbon Fiber Reinforced Polyamide 6 Composites Fabricated by Self-Resistance Electric Heating Technology"

_polymers, 2023, doi:10.3390/polym15081914_

Round 1
Reviewer 1 Report
Next-generation composite manufacturing processes are needed to overcome several limitations of conventional manufacturing processes (e.g., high energy consumption). Carbon fiber reinforced plastic self-resistance electric (SRE) heating is conceived as an alternative to out-of-autoclave technology due to its characteristics of uniform heating, fast heating/cooling, low energy consumption, and low equipment investment. The reviewed paper (manuscript ID: polymers-2326950, titled: Optimization of process parameters for carbon fiber reinforced polyamide 6 composites fabricated by self-resistance electric heating technology) presents results the results of the influence of process parameters on the quality of impregnation and the mechanical properties of new CF/PA 6 composite laminates with a focus on optimizing a set of process parameters. I have found the paper to be interesting. I appreciate the effort that the author has put in performing this study.
I have no objection to the work according to the substance, but please correct the incorrect numbers of the figures in the text from line 342 to the end of the article.
In general, what is the reproducibility of those experiments?
Reviewer 2 Report
The article "Optimization of process parameters for carbon fiber reinforced polyamide 6 composites fabricated by self-resistance electric heating technology" discussed the mechanical properties of CF/PA6 composite laminates manufactured by compression moulding with self heating technology. The topic is interesting and fall within the scopes of Polymers. Motivation is clear. Although sufficient experimental works were conducted and presented, the authors are suggested to improve the manuscript by considering the following suggestions:
1) line 36, Check the format of citation numbers (5 and 11).
2) Minor grammatical/style errors were detected.
line 112, lines 157-158 etc
3) please use a consistent format for space between numerical value and unit symbol throughout the manuscript, Inconsistency was detected, for example;" it was selected as 5min, 10min, 15min and 20 min".
4) Fig 3. T1-T6 were defined in 2.5.1. but Fig 3 was presented in 2.4.
5) Temperature measurement (Fig 3), please provide the number of repetitive measurement.
6) lines 144-146, please confirm the ref 34 is related to "the forming pressure was selected as 1.0 MPa, 1.5 MPa, 2.0 MPa and 2.5 MPa. As for impregnation time, it was selected as 5min, 10min, 15min and 20 min, based on references [33-34]".
7) Please further explain the orthogonal experimental plan, especially the selection of the lvl of processing temperature. Currently one 3- level factor and two 4-level factors were investigated.
8) line 196, Please explain how the samples were cut.
9) lines 208, 214. please provide the h, inherent thickness of laminates.
10) Lines 371, please further explain "the crystals grow gradually from the carbon fiber to the surrounding, which weakens the impact resistance of the matrix. Furthermore, due to the tight arrangement of grains around the fiber, the load transfer performance between the fiber and the matrix is also weakened."
11) unable to locate Fig 15 and Fig 1.
12) line 221, The input currents required for the three forming temperatures (250 ℃, 260 ℃, and 221 270 ℃) applied in this study are shown in Figure 4(a). -> cant find the above in Fig 4.
Please check the numbering of all figures.
Round 2
Reviewer 2 Report
the authors have revised the manuscript according to the reviewer's comment